# Leaf Litter Breakdown and Soil Microbes in *Catalpa bungei* Plantations in Response to Various Fertilization Regimes

Zhuizhui Guan [1], Tianxiao Chen [1], Dong Chen [1], Yizeng Lu [2], Qingjun Han [2], Ningning Li [2], Wenjun Ma [3], Junhui Wang [3], Yan Su [1], Jiyue Li [1], Quan Qiu [1,*] and Qian He [1,*]

1   Guangdong Key Laboratory for Innovative Development and Utilization of Forest Plant Germplasm, College of Forestry and Landscape Architecture, South China Agricultural University, Guangzhou 510642, China; 20201159005@stu.scau.edu.cn (Z.G.); 20203154003@stu.scau.edu.cn (T.C.); cd1359535346@stu.scau.edu.cn (D.C.); suyan@scau.edu.cn (Y.S.); ljyue@scau.edu.cn (J.L.)
2   Shandong Provincial Center of Forest and Grass Germplasm Resources, Institute of Resource Cultivation, Jinan 250102, China; luyizeng@126.com (Y.L.); 15064112007@163.com (Q.H.); 13953117854@163.com (N.L.)
3   Research Institute of Forestry, Chinese Academy of Forestry, Beijing 100091, China; mwjlx@sina.com (W.M.); wangjh808@sina.com (J.W.)
*   Correspondence: qqiu@scau.edu.cn (Q.Q.); heqian@scau.edu.cn (Q.H.)

**Abstract:** Litter decomposition propels the geochemical cycle by returning nutrients to soil. Soil microbial communities play an important role during litter breakdown wherein various fertilization regimes are conducted. In this study, we carried out a five-year fertilization experiment in a young *Catalpa bungei* plantation in northern China. The fertilization strategies employed mainly included the integration of water and fertilizer (WF), hole fertilization (HF), and no fertilization (CK) as a control. We tracked the decomposition dynamics of leaf litter and identified the major microbial communities involved in litter breakdown for each fertilization regime. The results showed that fertilization increased the biomass and C content of leaf litter, and the C storage in the HF forest was higher than that in the WF forest. Fertilization significantly decreased leaf litter decomposition and nutrient release and prolonged the duration of breakdown. The breakdown of litter in the WF stand was slower than that in the HF stand, but the diversities of bacteria and fungi were higher in the WF soil. The community structures of bacteria and fungi in the WF soil showed obvious differences compared to those in the CK and HF soils. Fertilization strengthened competitive relationships but decreased cooperative interaction among microbes. The abundances of saprophytic fungi and decomposing bacteria in the WF soil were lower than those in the HF soil. The key flora, including *Arthrobacter* and *Neocosmospora*, regulated litter breakdown in the HF and WF forests. In addition, *Arthrobacter*, *Filobasidium*, and *Coprinopsis* were mainly involved in the decomposition process in the nonfertilized forests. Thus, studying the biomass and initial quality of litter treated with different fertilization measures and exploring the characteristics of nutrient release during litter decomposition are both of significant value with regard to deepening understanding of the effects of different fertilization methods on litter breakdown and their associated response mechanisms.

**Keywords:** *Catalpa bungei*; soil microbial community; leaf litter; decomposition rate; fertilization regimes; integration of water and fertilizer





## 1. Introduction

Plants draw nutrients from the soil to grow and then release these nutrients as litter back into the soil [1–3]. Litter decomposition drives nutrient cycling in all vegetative ecosystems [4–6]. The decomposition of litter is largely governed by climatic conditions [5,7], insects [8], the type of ecosystem in which the process will occur [2], whether the surrounding trees are evergreen or deciduous [9], and litter quality (e.g., lignin and cellulose) [6,10]. In addition, microbes can engage in the process of breaking down litter [11–15]. The authors of [16] indicated that the mass loss of leaf litter in *Pinus tabuliformis* plantations was

positively correlated with fungal diversity. Similarly, the authors of [17] also found that the breakdown rate of litter showed positive correlations with the diversity and richness of soil bacteria and fungi. However, in [18], it was observed that the decomposition rate of litter was negatively correlated with the diversity and richness of bacterial and fungal communities. These studies highlight the crucial role that microbes play in the decomposition of litter, but the pivotal bacterial and fungal flora that are involved in this process are poorly understood.

The decomposition of litter is also affected by different fertilization regimes [19–21]. Currently, the number of reports on how fertilization affects litter breakdown and microbial groups is increasing [22–25]. Such studies have mainly focused on the response of litter decomposition to conventional fertilization (e.g., hole application). For example, the authors of [26] found that fertilization (e.g., N input) reduced the microbial community and lowered enzyme activity in the analyzed soil, which further decreased the decomposition rate and net N release of litter. The authors of [27] also concluded that N deposition generally suppressed the process of litter decomposition, and other nutrient inputs enhanced the inhibitory effects of N input on litter breakdown. However, the authors of [28] concluded that increasing the amount of N fertilizer applied helped improve the breakdown process by decreasing the C/N ratio of litter. It has also been observed that increasing rates of long-term N input can increase litter decomposition in a semiarid grassland [29]. In addition, according to the authors of [22], N and P input improved the richness of microbes and accelerated the decomposition of litter. Thus, the effect of nutrient input on litter breakdown may be promotive [30,31] or inhibitive [32,33].

The integration of water and fertilizer can improve soil conditions, plant yield and quality, and boost the water and fertilizer efficiency of crops (by approx. 30%–60%) compared to conventional fertilization in agricultural ecosystems [34]. Nevertheless, this technology is rarely utilized in forestry ecosystems. The authors of [35] observed that drip fertigation significantly increased biomass and carbon storage in a poplar plantation. The 11-year study presented in [36] indicated that the adoption of fertigation increased productivity and ensured higher efficiency of the two most critical inputs, i.e., water and nutrients, in arecanut production. These reports concentrate solely on the growth, productivity, and nutrient uptake of trees in response to this technique. Few studies have deeply examined how the integration of water and fertilizer affects nutrition return (e.g., litter decomposition) by impacting soil microbial communities. Thus, the relationship among fertigation, soil microbes, and litter breakdown needs immediate clarification.

*Catalpa bungei*, a valuable ornamental wood, grows in northern China. We carried out a five-year fertilization trial that included the integration of water and fertilizer, hole fertilization, and no fertilization as a control. The decomposition of leaf litter and soil microbial changes under various fertilization methods were monitored. We attend to address the following questions in our study. First, do the two fertilization regimes promote, inhibit, or fail to affect leaf litter decomposition compared to no fertilization? (Additionally, does the effect of the integration of water and fertilizer differ from that of hole fertilization?) Second, how does fertilization affect leaf litter breakdown by acting on soil microbial communities? Finally, what are the key microbial populations that influence leaf litter breakdown in different forests?

## 2. Materials and Methods

### 2.1. Study Plot and Tree Species

The study plot is located at the Jujube Preservation Warehouse in Zhangqiu District, Jinan City, Shandong Province, China (36°25′–37°09′ N, 117°10′–117°35′ E). The region has a moderate monsoon climate with average annual temperatures of 12.8 °C, a highest monthly temperature of 27.2 °C (July), and a lowest monthly temperature of −3.2 °C (January). The area's average annual rainfall value is 600.8 mm. There are a total of 2647.6 h of sunlight and 192 days without frost, annually. The growing season of *C. bungei* lasts 130–150 days, from May to September. The wet season in Shandong Province is from May to

September, and the dry season spans the remaining months. Generally, high temperatures occur in the wet season, while low temperatures occur in the dry season.

In March 2017, the pure *C. bungei* plantations were planted with a 2-year-old clone ("9-1") in a 3 × 4 m planting grid. A total of 18 plots (ca. 0.8 ha) were established, and 45 trees were planted in each plot (5 rows × 9 columns). The plantations had not been tended since planting. The mean tree height (ca. 4.2 m) and diameter at breast height (ca. 4.0 cm) were measured immediately after planting. The chemical properties of the soil, including its pH (7.67), content of soil organic matter (19.64 g/kg), total N (0.91 g/kg), total P (0.53 g/kg), total K (16.7 g/kg), alkeline-N (81.88 mg/kg), available P (32.10 mg/kg), and available K (176.82 mg/kg), were assessed. A plot diagram is displayed in Figure S1.

*2.2. Fertilization*

In early May 2018, we conducted a split-plot experiment and randomly selected 9 plots for fertilization. The average size of each plot was approximately 384 m$^2$. In our study, hole fertilization refers to the application of the optimum amount of fertilization based on previous fertilization experience in *C. bungei* plantations. The water–fertilizer integration technique commonly used in agricultural ecosystems was also adopted. Thus, two fertilization methods were developed, hole fertilization (HF) and the integration of water and fertilizer (WF), while a no fertilization (CK) scheme was applied as a control. Each treatment was distributed randomly in three plots, and the plots were spaced 6–8 m apart.

In the past three years, several fertilization amounts for the fertilization of *C. bungei* have been tested. In addition, the optimum regime for hole fertilization, i.e., the application of N (24 g/tree), $P_2O_5$ (8 g/tree), and $K_2O$ (16 g/tree), was selected for *C. bungei*-fertilized plantations and was never published. In this study, another fertilization scheme, i.e., integration of water and fertilizer, which is widely used in agricultural ecosystems, was considered, consisting of the use of the same amount of fertilizer as that used in hole fertilization.

The first fertilization scheme was administered in May 2018. Three fertilizers, N (24 g/tree), $P_2O_5$ (8 g/tree), and $K_2O$ (16 g/tree), were employed in the studied HF and WF forests. Starting in the second year, the total amount of each fertilizer applied for each year increased by 20% compared to the year before. In May of each year, the HF scheme constructed the application of all fertilizers at once. A hole 20 cm in diameter and 30 cm in depth was excavated to the south and north of each tree. Then, the fertilizers were equally divided into 2 parts and placed into the holes. The annual fertilizers of the WF scheme were divided into 12 portions that were distributed equally. Starting on 1 May, we fertilized the subjects once every 10 days. Using an intelligent drip irrigation system (HN-BXE, Huinong Automation Corporation, Beijing, China), the fertilizers were correctly delivered close to the roots after being dissolved in 1000 L of water. Regarding the HF and WF schemes applied in this study, potassium sulfate was employed as the potassium fertilizer and urea as the nitrogen fertilizer. Calcium superphosphate was utilized as a phosphate fertilizer for HF, whereas ammonium dihydrogen phosphate was used for WF. All fertilizers were purchased on the Taobao website. The local meteorological data (e.g., temperature and precipitation) are shown in Figure S2.

*2.3. Leaf Litter Collection and Decomposition*

The litter collector covered a surface area of about 1 m$^2$ (with 1 mm mesh spacing) and was constructed with nylon mesh. The vertical distance between the bottom of the collector and the ground was approximately 0.5 m. Five collectors were arranged randomly in each plot (see Figure S1) [3]. All *C. bungei* leaves naturally fall off before December. No leaves were observed from December to April. Thus, the leaf litter was gathered at the conclusion of each month from May to November 2021. The dry weight (i.e., biomass) and carbon content of the leaf litter were measured.

The breakdown dynamics of leaf litter were monitored using the decomposition bag approach. In each plot, we gathered samples of fresh litter of the same size at the end of May 2021. Then, the litter was air-dried after using deionized water to remove any sand.

A 25 × 15 cm nylon bag with 0.425 mm mesh spacing was used to collect 20 g of litter after the litter was air-dried. Fifteen bags were distributed at random in each plot. Some litter needed to be placed on top of the bag, which was in contact with the soil. Three bags from each plot were retrieved at approximately 60-day intervals beginning on 30 July 2021. The decomposition of leaf litter occurred throughout 0-, 60-, 120-, 180-, 240-, and 300-day time periods (30 May, 30 July, 30 September, 30 November, 30 January, and 30 March, respectively). The litter was cleaned using deionized water, dried at a temperature of 75 °C over 48 hours or until a constant weight was obtained, and then weighed when the bags were collected. The concentrations of total N, P, K, Ca, Mg, organic carbon (C), C:N, C:P, and N:P in leaf litter were measured.

## 2.4. Soil Sampling

The soil samples were gathered in free areas of the forest immediately after all the bags were recollected. Five topsoil samples from each plot (i.e., 0–20 cm) were chosen at random, mixed evenly, sieved (using a sieve with a mesh size of 60), and then split into two soil samples. Some soil samples were air-dried to determine their chemical properties, while other soil samples were preserved at −80 °C for microbial analysis. Information on soil properties is shown in Table S1.

## 2.5. Elemental Analysis of Leaf Litter and Soil

The pH, organic carbon (SOC), organic matter (SOM), alkeline-nitrogen (AN), available phosphorous (AP), and available potassium (AK) in the soil were determined. The total nitrogen (TN), total phosphorous (TP), and total potassium (TK) in the soil and leaf litter were measured. The concentrations of Ca, Mg, and C in the leaf litter were measured.

The determination of all properties in the litter and soil was carried out according to the methods reported in [37]. Soil pH was measured using the potentiometric approach (1:2.5 soil-to-water ratio). The SOM of soil and leaf litter was measured using the volumetric potassium dichromate technique. SOC was calculated by dividing SOM by 1.724. The TN of soil and leaf litter was measured using the acid digestion–indophenol blue colorimetric method. The TP of soil and leaf litter was measured using the acid digestion–molybdenum antimony resistance colorimetric method. The TK of soil and leaf litter was measured using acid digestion–flame atomic absorption. The determination of soil AK was performed by referencing TK. Soil AN was measured using a colorimetric method consisting of indophenol blue and potassium chloride leaching. Soil AP was measured using hydrochloric acid/ammonium fluoride/sodium bicarbonate leaching–molybdenum antimony anti-colorimetry. The amounts of Ca and Mg in leaf litter were calculated using the dry ash technique–dilute hydrochloric acid dissolution method.

## 2.6. Molecular Analysis of Soil Microbes

The total genomic DNA of the samples was extracted using the CTAB method. DNA concentration and purity were monitored on 1% agarose gels. DNA was diluted with sterile water to 1 ng/μL depending on the concentration. In order to amplify the V3–V4 region of the bacterial 16S rRNA gene, the primers 341F and 806R (5′–CCTAYGGGRBGCASCAG–3′ and 5′–GGACTACNNGGGTATCTAAT–3′) were used [38]. The fungal internal transcribed spacer (ITS) was amplified using the primers ITS1-1F-F (5′–CTTGGTCATTTAGAGGAAGTAA–3′) and ITS1-1F-R (5′–GCTGCGTTCTTCATCGATGC–3′) (Bokulich et al., 2018). A total of 15 L of Phusion High-Fidelity PCR Master Mix was used for all PCR experiments (T100PCR, Bio-Rad, Hercules, CA, USA). The experiment incorporated 1 μL each of forward primers (2 μM/μL) and reverse primers (2 μM/μL) and about 10 ng of template DNA. Thermal cycling consisted of initial denaturation at 98 °C for 1 min, followed by 30 cycles of denaturation at 98 °C for 10 s, annealing at 50 °C for 30 s, and elongation at 72 °C for 30 s and then 72 °C for 5 min. PCR products were separated on a 2% agarose/1× TAE gel and purified with a Qiagen Gel Extraction Kit (Qiagen, Hilden, Germany).

According to the manufacturer's instructions, sequencing libraries were created using the TruSeq DNA PCR-Free Sample Preparation Kit (Illumina, San Diego, CA, USA) and index codes were added. The library's quality was assessed on the Qubit 2.0 Fluorometer (Life Technologies, Carlsbad, CA, USA). Finally, 250 bp paired-end reads were produced after the library was sequenced on an Illumina NovaSeq platform (Novaseq 6000, Illumina, San Diego, CA, USA). The number of sequences reached 50,000.

The analysis was conducted by following the "Atacama soil microbiome tutorial" of Qiime2docs along with customized program scripts (https://docs.qiime2.org/2019.1/ (accessed on 3 February 2022)). Briefly, the qiime tools import program was used to import the raw data FASTQ files into a format that was compatible with the QIIME2 system. Demultiplexed sequences from each sample were quality-filtered and trimmed, de-noised, and merged; then, the chimeric sequences were identified and removed using the QIIME2 dada2 plugin to obtain a feature table of amplicon sequence variants (ASV) [39]. The QIIME2 feature-classifier plugin was then used to align ASV sequences to a pre-trained GREENGENES 13_8 99% database (trimmed to the V3V4 region bound by the 341F/806R primer pair or ITS region bound by the ITS1-1F-F/ITS1-1F-R primer pair) to generate a taxonomy table. The default UNITE database version of ITS was 8.2. Any contaminative mitochondrial and chloroplast sequences were filtered using the QIIME2 feature-table plugin.

### 2.7. Data Analysis

Two-way ANOVA was used to explore the effects of fertilization, month, and their interactions on the biomass and C content of leaf litter. Tukey's HSD was used to examine differences between fertilization treatment or month.

The residual mass rate of leaf litter was calculated using the method presented in [40]. The formula is as follows:

$$\mathrm{RM} = \frac{M_t}{M_0} \times 100\%$$

where the RM is the residual mass rate of leaf litter, $M_t$ is the mass of leaf litter at decomposition time t, and $M_0$ is the initial mass of leaf litter.

The decomposition constant of leaf litter was calculated using the Olson model [41]. The model is as follows:

$$\frac{M_t}{M_0} = a \times e^{-(k \times t)}$$

where $a$ is the correction parameter, $k$ is the decomposition constant, t is the decomposition time, and e is the constant (i.e., 2.71828).

The following formulas were used to determine the time needed for leaf litter to decay by 50% and 95%:

$$t_{50\%} = -\ln\left(\frac{0.5}{k}\right)$$

$$t_{95\%} = -\ln\left(\frac{0.05}{k}\right)$$

The element release rate of leaf litter was calculated using the ratio of the amount of released elements to the initial elemental content of leaf litter. The formula for the element release rate of leaf litter is as follows:

$$E = \frac{(E_0 - E_t)}{E_0} \times 100\%$$

where E is the element release rate of leaf litter, $E_0$ is the product of the initial biomass and element concentration of leaf litter, and $E_t$ is the product of biomass and element concentration of leaf litter at decomposition time t.

The analyses of the mass residual rate, element concentration, and element release rate in leaf litter were similar to that of the biomass of leaf litter. Tukey's HSD was used to examine differences between fertilization or decomposition time.

The chemical properties of soil were analyzed by one-way ANOVA with Tukey's HSD for post hoc tests. The QIIME2 core-diversity plugin was used to compute microbial diversity. The alpha diversity indices (e.g., OTU, Chao1, Shannon, and Simpson's indices) were calculated to express the degree of microbiological diversity within a specific sample. The number of OTUs indicated the richness of a guild. The relative abundance of each guild was calculated by dividing the number of sequences of a specific guild by the total number of sequences [42]. The structural variance of microbial communities among samples was investigated via beta diversity distance metrics (e.g., Bray–Curtis) and then depicted via nonmetric multidimensional scaling (NMDS) [43]. Redundancy analysis (RDA) was performed to reveal the relationships between environmental factors and microbial communities based on the relative abundances of microbial species at different genus levels using the R package "vegan" [44]. The FUNGuild tool was used to assign each OTU to an ecological guild to ascertain how different fertilization regimes affected the fungal function guilds [42,45,46]. Relationships between soil chemistry, microbial diversity, and stoichiometry of leaf litter were examined via the Spearman approach. Spearman analysis was also used to determine the interaction between microbial species based on relative abundances. The "igraph" package in R was used to create a relevant network diagram for soil microbes. The network diagram was used to search for species that were mutually antagonistic or synergistic.

## 3. Results

### 3.1. Biomass and C Content of Leaf Litter

Fertilization increased the biomass and C content of the analyzed leaf litter. HF increased biomass by 9.0% and 15.1% and C storage by 10.0% and 16.3% compared to the WF and CK treatments, respectively (Table S2). The biomass change in the leaf litter was comparatively stable from May to August, accounting for 1.83%–2.39% of the annual share of litter. The biomass of leaf litter progressively increased after September and accounted for 4.42% (September), 31.64% (October), and 56.21% (November) of the annual litter shares (Figure 1a). The C concentrations of leaf litter in October and November were much higher than those from May to September and followed a similar pattern to that of biomass (Figure 1b).

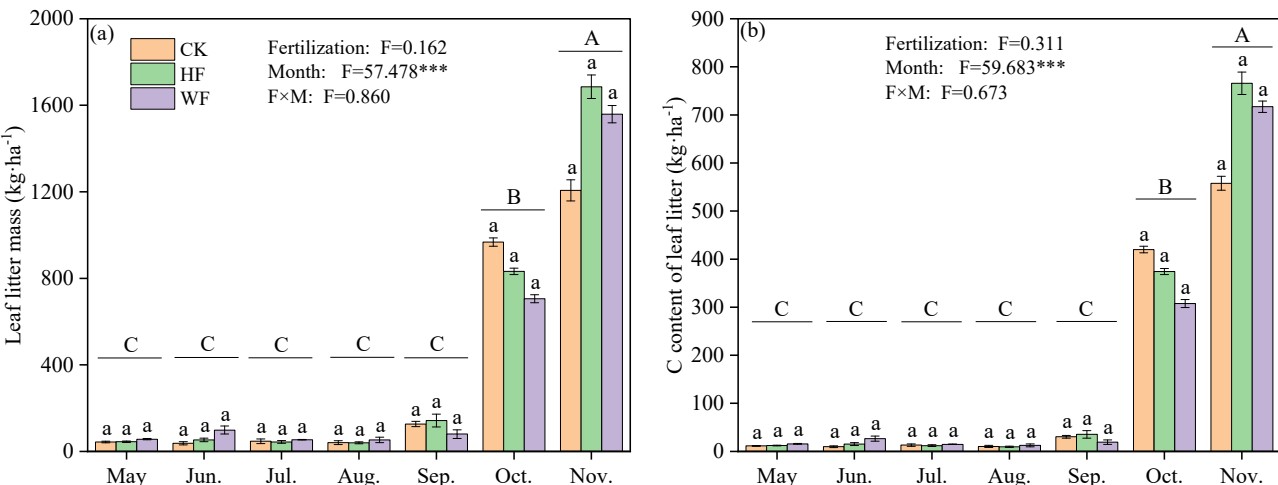

**Figure 1.** Monthly dynamics of mass (**a**) and C content (**b**) of leaf litter. Different capital letters indicate significant differences between months ($p < 0.001$, Tukey's HSD). Different lowercase letters indicate significant differences between fertilization treatments ($p < 0.001$, Tukey's HSD). CK, no fertilization; HF, hole fertilization; and WF, integration of water and fertilizer. ***, $p < 0.001$.

### 3.2. Decomposition of Leaf Litter

The biomass of leaf litter decreased gradually in the three treatments. Leaf litter decomposed quickly before 60 days and then slowly declined after 120 days (Figure 2). The decomposition rate of leaf litter was dramatically altered by fertilization, which presented slower breakdown than that effected by the CK treatment. The breakdown rate of litter via the WF treatment was slower than that afforded by the HF treatment. Fertilization significantly reduced the $k$ value but increased $t_{50\%}$ and $t_{95\%}$ (Table 1; Table S3). The decomposition of 50% of the leaf litter via the HF and WF treatments required 0.04 and 0.10 more years, respectively, than the CK treatment. It took longer to decompose 95% of the leaf litter in the HF and WF treatments (0.18 and 0.43 years, respectively) than in the CK treatment.

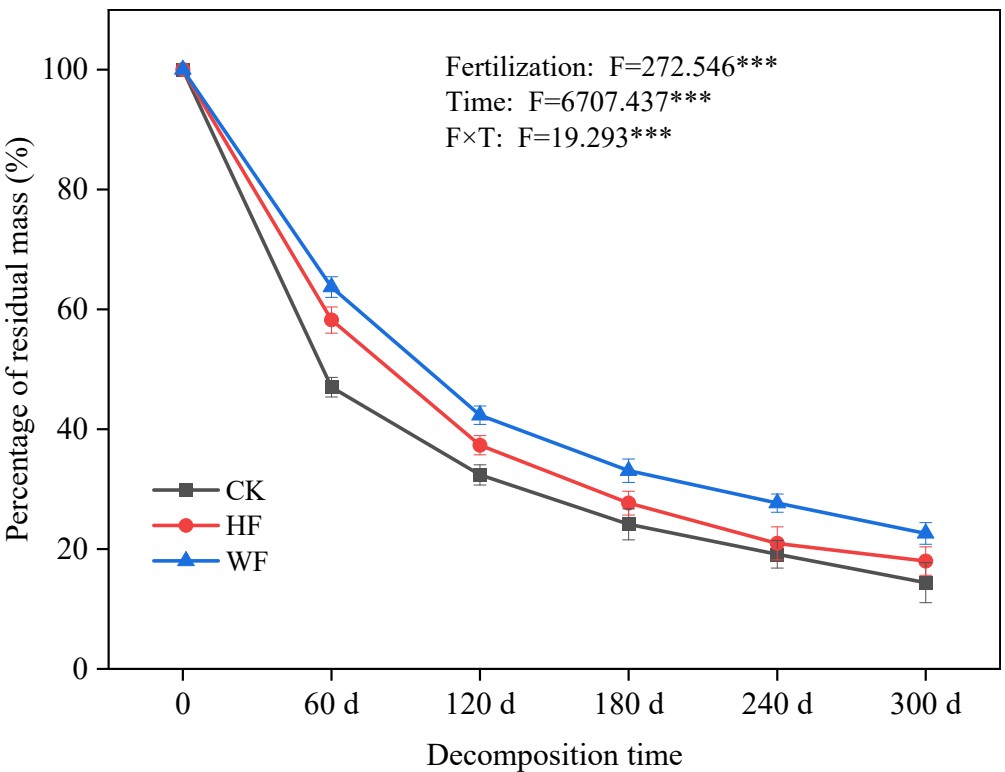

**Figure 2.** Decomposition dynamics of leaf litter under different fertilization regimes: CK, no fertilization; HF, hole fertilization; and WF, integration of water and fertilizer. *** $p < 0.001$.

**Table 1.** Olson exponential decay models in three fertilization regimes.

| Fertilization | Litter Number | $a$ | $k$ | 95% Confidence Interval Lower Bound | Upper Bound | Equation | $R^2$ | $t_{50\%}$ (a) | $t_{95\%}$ (a) |
|---|---|---|---|---|---|---|---|---|---|
| CK | CK-1 | 94.014 | 2.862 | 2.363 | 3.362 | $Y = 94.014 \times \text{EXP}(-2.862 \times t)$ | 0.935 | 0.242 | 1.047 |
| | CK-2 | 95.058 | 2.984 | 2.545 | 3.422 | $Y = 95.058 \times \text{EXP}(-2.984 \times t)$ | 0.955 | 0.232 | 1.004 |
| | CK-3 | 95.392 | 3.176 | 2.712 | 3.640 | $Y = 95.392 \times \text{EXP}(-3.176 \times t)$ | 0.957 | 0.218 | 0.943 |
| HF | HF-1 | 97.497 | 2.630 | 2.375 | 2.886 | $Y = 97.497 \times \text{EXP}(-2.630 \times t)$ | 0.979 | 0.264 | 1.139 |
| | HF-2 | 96.181 | 2.498 | 2.230 | 2.766 | $Y = 96.181 \times \text{EXP}(-2.498 \times t)$ | 0.973 | 0.277 | 1.199 |
| | HF-3 | 95.935 | 2.500 | 2.222 | 2.777 | $Y = 95.935 \times \text{EXP}(-2.500 \times t)$ | 0.971 | 0.277 | 1.198 |
| WF | WF-1 | 95.526 | 2.061 | 1.810 | 2.312 | $Y = 95.526 \times \text{EXP}(-2.061 \times t)$ | 0.961 | 0.336 | 1.454 |
| | WF-2 | 96.688 | 2.096 | 1.895 | 2.297 | $Y = 96.688 \times \text{EXP}(-2.096 \times t)$ | 0.976 | 0.331 | 1.429 |
| | WF-3 | 96.304 | 2.135 | 1.928 | 2.343 | $Y = 96.304 \times \text{EXP}(-2.135 \times t)$ | 0.976 | 0.325 | 1.403 |

Note: $Y$ = percentage of residual mass of leaf litter, $Y = M_t/M_0$; $M_0$ = initial mass of leaf litter; $M_t$ = mass of leaf litter at decomposition time t; $a$ = correction parameter; $k$ = decomposition constant; $t_{50\%}$ = the duration for decomposing 50% of leaf litter; and $t_{95\%}$ = the duration for decomposing 95% of leaf litter. CK, no fertilization; HF, hole fertilization; and WF, integration of water and fertilizer.

### 3.3. Stoichiometry of Leaf Litter

The N and P content of leaf litter declined quickly from 0 to 60 days and showed a fluctuating reduction from 120 to 300 days in each treatment (Figure 3a,b). The N and P content had decreased by 25.2%–39.6% and 49.1%–57.3%, respectively, by the end of the trial compared to the beginning in all treatments. The K content fell rapidly from 0 to 120 days and, subsequently, fluctuated from 120 to 300 days (Figure 3c). The K content had dropped by 75.1%–83.4% by the conclusion of the experiment compared to that at the beginning. The Ca content increased quickly before 60 days and then decreased from 60 to 120 days (Figure 3d). The Ca content increased by 13.3%–31.3% from the beginning to the end of the trial. The Mg content varied between 2.427 and 6.615 mg/g in all treatments (Figure 3e). The C content declined quickly over the first 120 days and then increased (Figure 3f). C:N, C:P, and N:P ratios fluctuated upwardly over time (Figure S3).

### 3.4. Elemental Release from Leaf Litter

The average release rates of N, P, K, Ca, Mg, and C content from leaf litter were significantly reduced by fertilization (Table S4). The release rates of N, P, K, and C through WF were obviously lower than those afforded via HF (Table S4). N, P, K, and C had a rapid release over the first 60 days (exceeding 55.8%) and then released more slowly in the three forests (Figure 4a–c,f). The Ca content accumulated from 0–60 days and then gradually released (Figure 4d). The release rate of Mg rapidly increased between 60 and 120 days and then slowly increased over time (Figure 4e).

### 3.5. Microbial Composition

Proteobacteria (30%–48%), Actinobacteria (21%–34%), Bacteroidetes (8%–14%), Firmicutes (6%–15%), and Acidobacteria (6%–11%) were the dominant phyla of bacteria in all soil samples (Figure 5a). Fertilization significantly reduced the quantity of Proteobacteria but increased the abundances of Bacteroidetes, Firmicutes, Acidobacteria, Gemmatimonadetes, and Verrucomicrobia (Table S5). The abundances of Actinobacteria, Bacteroidetes, Firmicutes, and Verrucomicrobia in the WF forest were slightly higher than those in the HF forest.

Ascomycota (61%–81%), Mortierellomycota (11%–24%), and Basidiomycota (6%–18%) were the most prevalent phyla of fungi in all soil samples (Figure 5b). Fertilization reduced the quantity of Ascomycota and Basidiomycota but increased the number of Chytridiomycota (Table S5). The abundances of Ascomycota, Mortierellomycota, Basidiomycota, and Chytridiomycota in the HF forest were slightly larger than those in the WF forest.

### 3.6. Microbial Diversity and Community

Fertilization increased the diversity of bacteria and fungi. The OTU, Chao1, Shannon, and Simpson indices of microbes in the WF forest were slightly higher than those in the HF forest (Table S6). The bacterial communities in the three forests were completely different (Figure 6a). The fungal communities of the CK and HF forests partially overlapped, indicating some similarities (Figure 6b). The fungal community in the WF forest was entirely different from that in the CK and HF forests.

### 3.7. Relationships between Microbial Diversity and Community, Litter Nutrients, Decomposition Constant (k), and Soil Properties

The decomposition constant (*k*) of the leaf litter was not correlated with microbial diversity (Figure S4). The bacterial Shannon index was positively associated with the SOM, SOC, and TP of soil and the N content and N/P ratio of leaf litter (Figure 7). The fungal Shannon index was only positively correlated with the Ca content of the leaf litter. The fungal Chao1 indices only correlated positively with the AK of soil. SOM and SOC were positively correlated with the TN and TP of soil as well as the N content of leaf litter.

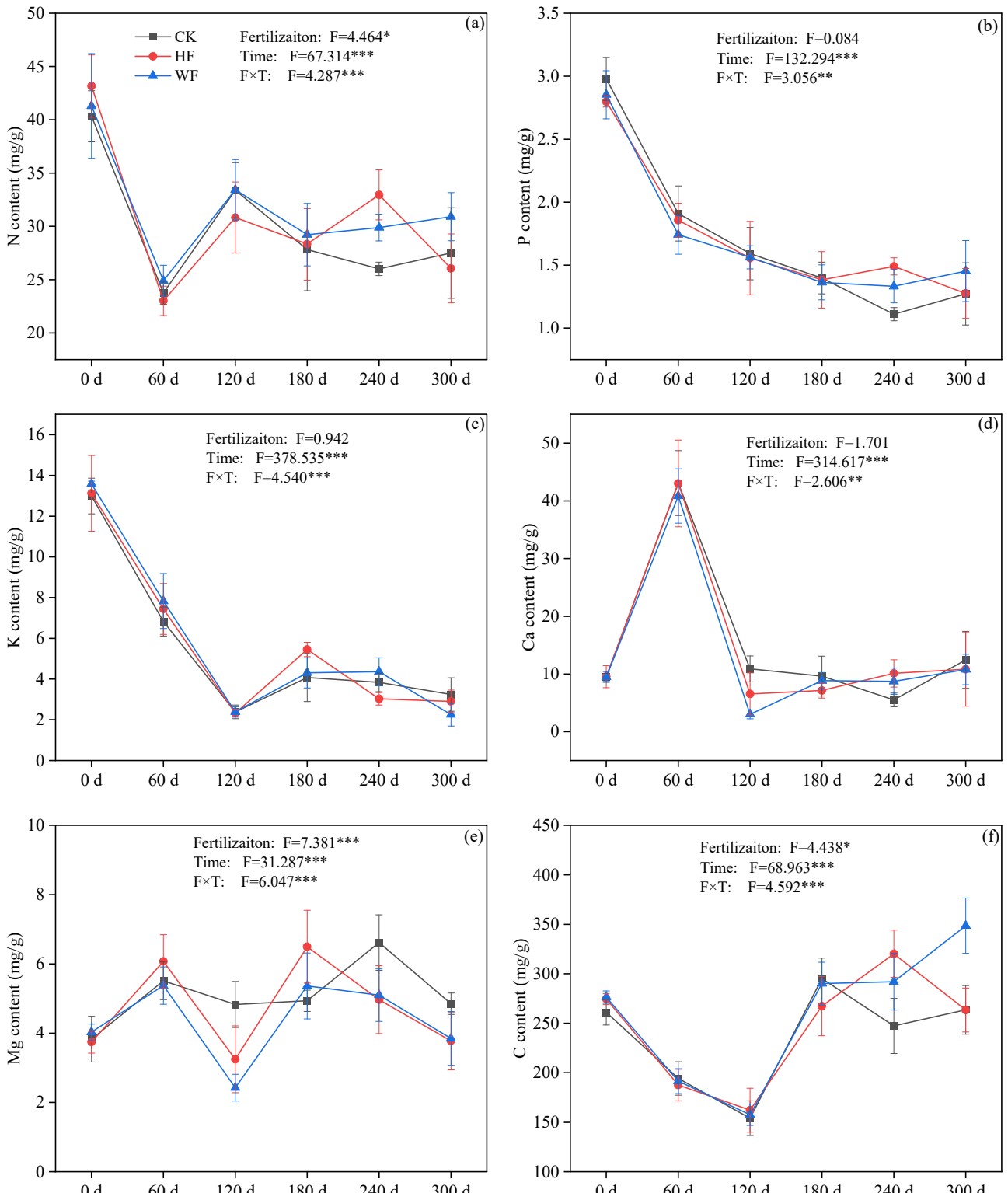

**Figure 3.** Dynamics of element concentrations in leaf litter under three fertilization regimes: (**a**), N content; (**b**), P content; (**c**), K content; (**d**), Ca content; (**e**), Mg content; and (**f**), C content. CK, no fertilization; HF, hole fertilization; and WF, integration of water and fertilizer. * $p < 0.05$, ** $p < 0.01$, and *** $p < 0.001$.

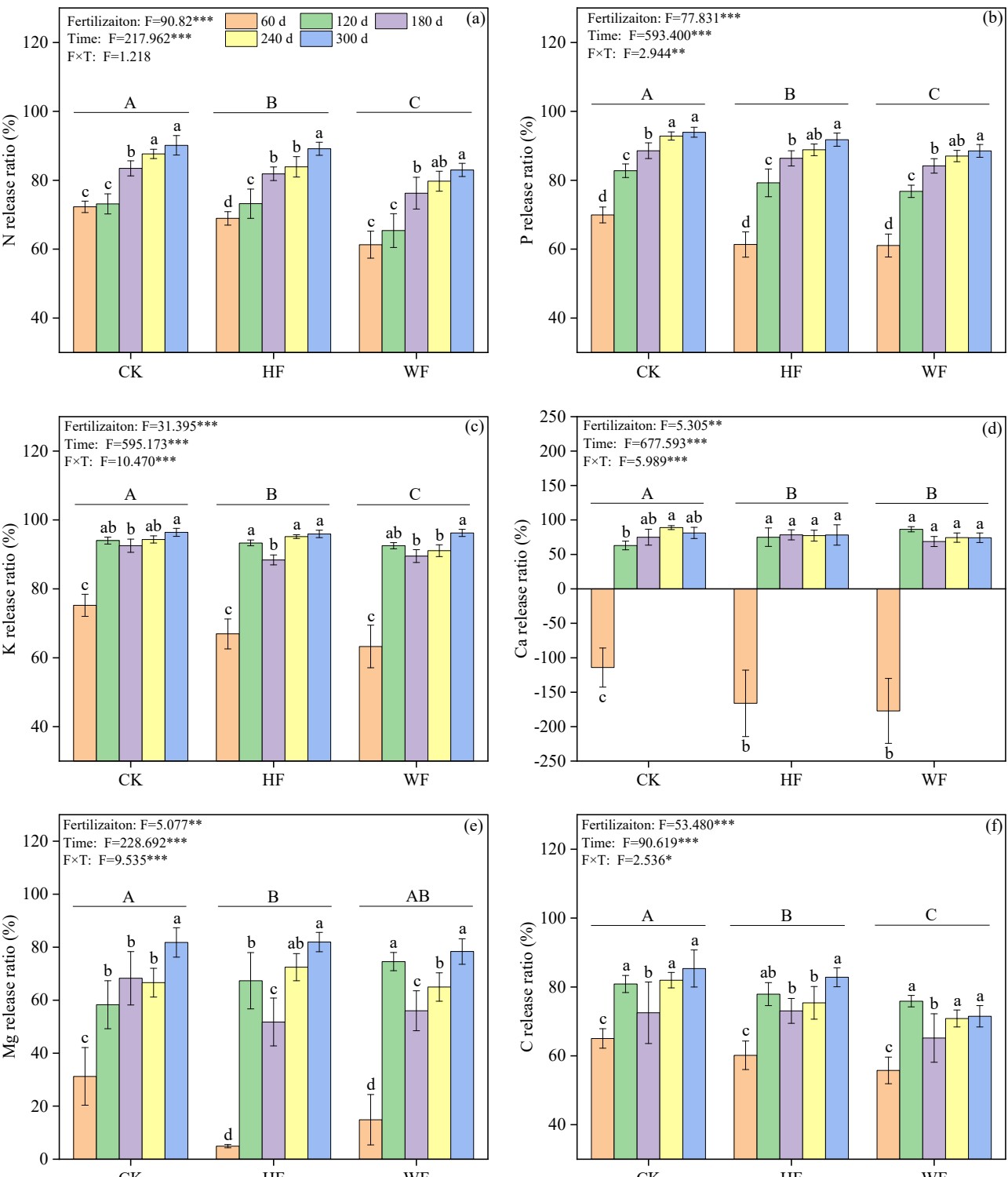

**Figure 4.** Elemental release ratio of leaf litter under three fertilization regimes: (**a**), N release ratio; (**b**), P release ratio; (**c**), K release ratio; (**d**), Ca release ratio; (**e**), Mg release ratio; and (**f**), C release ratio. CK, no fertilization; HF, hole fertilization; and WF, integration of water and fertilizer. Different capital letters indicate significant differences between fertilization ($p < 0.01$, Tukey's HSD). Different lowercase letters indicate significant differences between time ($p < 0.001$, Tukey's HSD). * $p < 0.05$, ** $p < 0.01$, and *** $p < 0.001$.

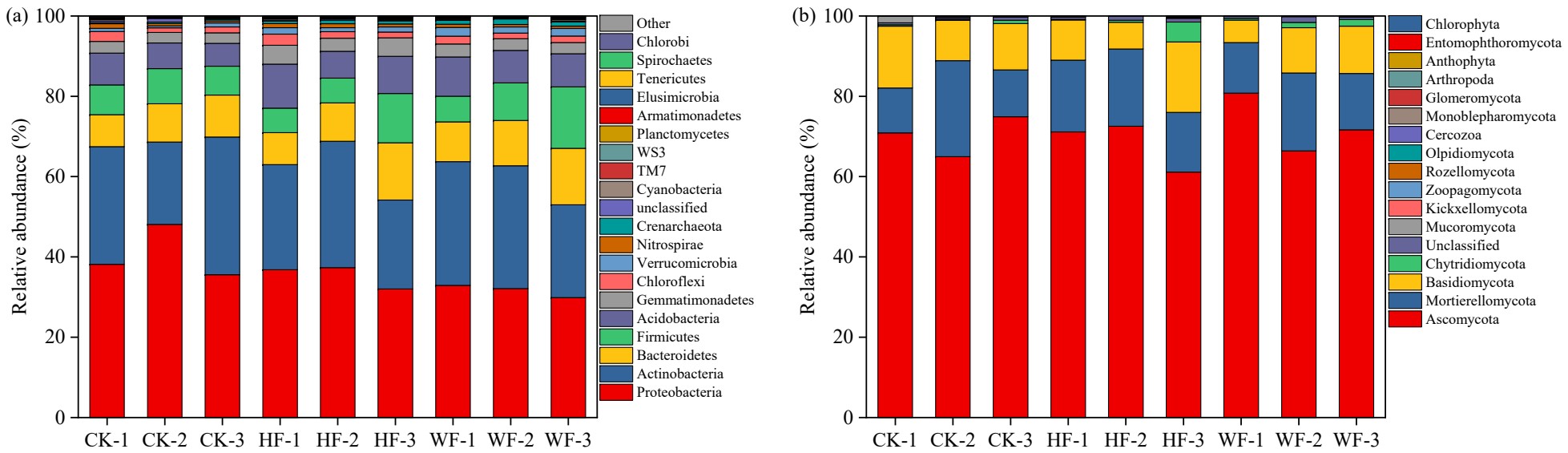

**Figure 5.** Relative abundance of microbes in the three treatments: (**a**) bacterial phyla and (**b**) fungal phyla. CK, no fertilization; HF, hole fertilization; and WF, integration of water and fertilizer.

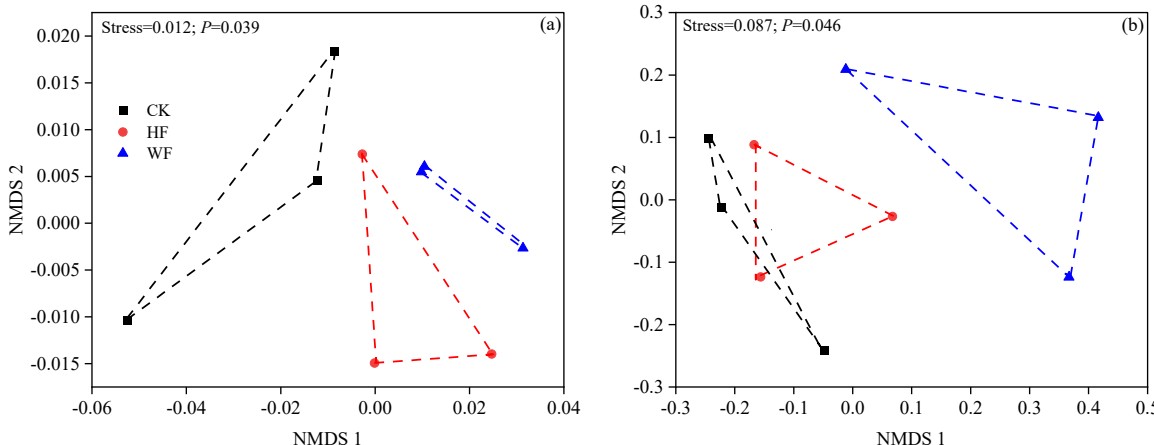

**Figure 6.** Nonmetric multidimensional scaling (NMDS) analysis based on Bray–Curtis dissimilarity values for microbial beta diversity: (**a**) bacteria and (**b**) fungi. CK, no fertilization; HF, hole fertilization; and WF, integration of water and fertilizer.

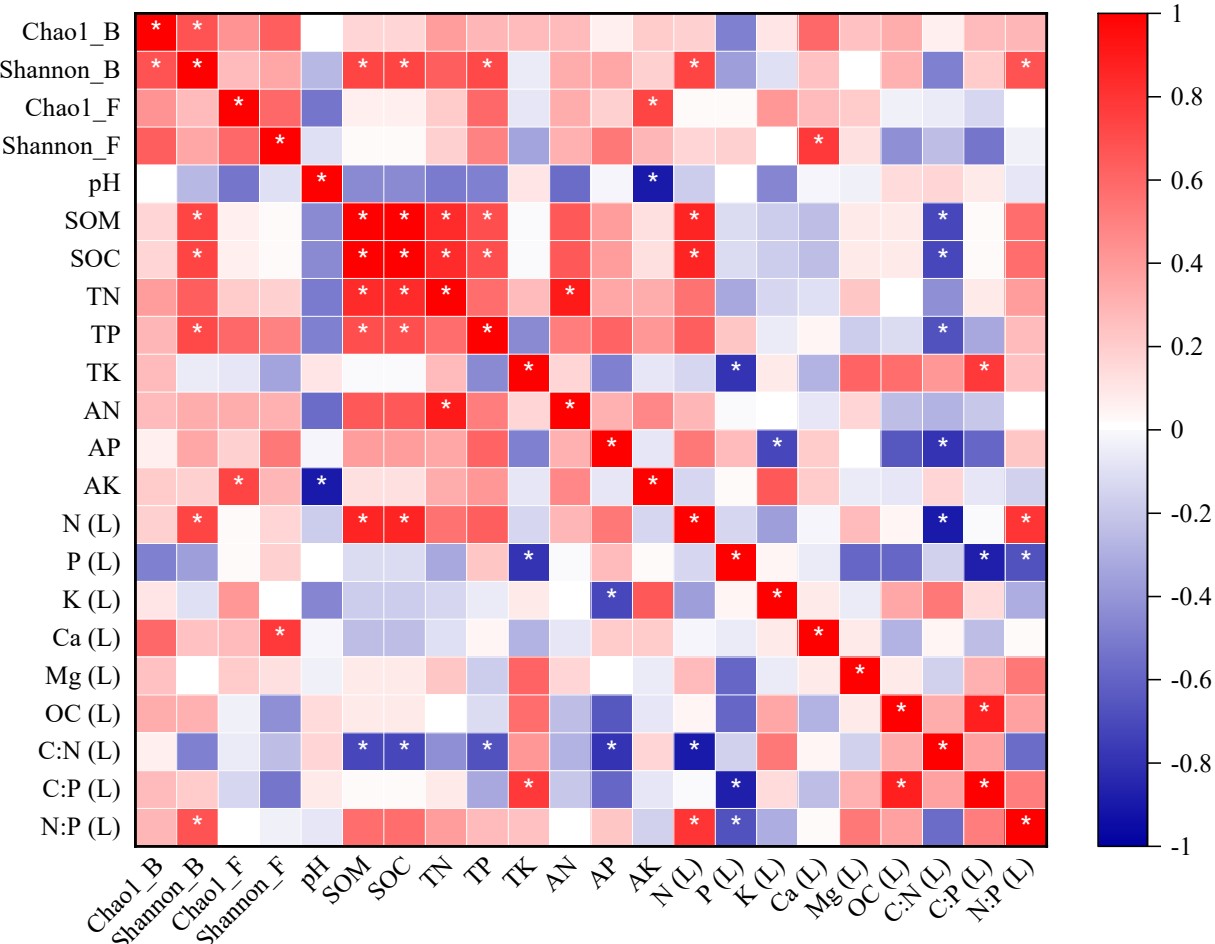

**Figure 7.** Relationships between microbial diversity, soil properties, and leaf litter chemometrics: Chao1_B, Chao1 index of bacteria; Shannon_B, Shannon index of bacteria; Chao1_F, Chao1 index of fungi; Shannon_F, Shannon index of fungi; SOM, soil organic matter; SOC, soil organic carbon; TN, total nitrogen; TP, total phosphorus; TK, total potassium; AN, alkeline–nitrogen; AP, available phosphorus; AK, available potassium; N(L), N content of leaf litter; P(L), P content of leaf litter; K(L), K content of leaf litter; Ca(L), Ca content of leaf litter; Mg(L), Mg content of leaf litter; OC(L), organic carbon content of leaf litter; C:N (L), C:N ratio of leaf litter; C:P(L), C:P ratio of leaf litter; and N:P(L), N:P ratio of leaf litter. * $p < 0.05$.

The studied bacterial communities were influenced by key elements, including the pH, SOM, SOC, and TN of the soil and the N, P, Mg, and C content and C:P and N:P ratios of leaf litter (Table S7; Figure 8a). The fungal communities were affected by many factors, such as the TN, TP, AN, and AP of the soil and the K, Ca, and C content of the leaf litter (Table S7; Figure 8b).

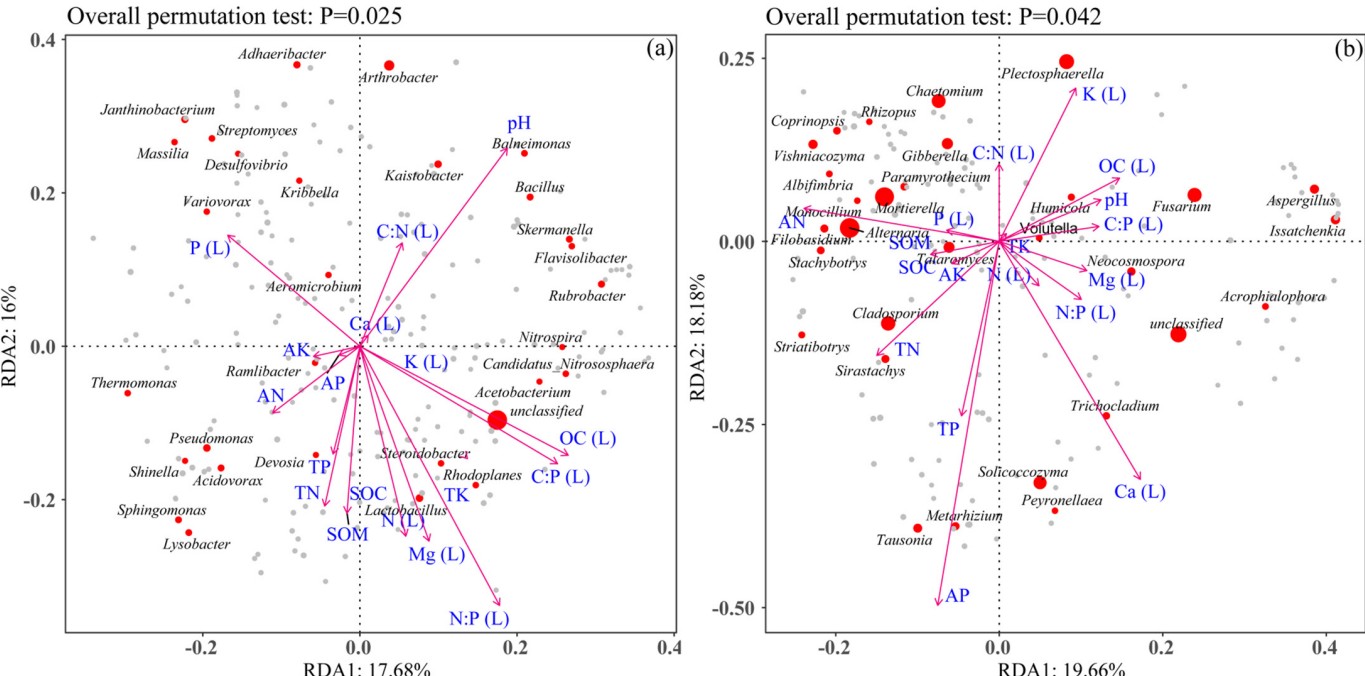

**Figure 8.** Redundancy analysis (RDA) between microbial communities (genus level) and soil properties and leaf litter stoichiometry; (**a**), bacteria; (**b**), fungi; SOM, soil organic matter; SOC, soil organic carbon; TN, total nitrogen; TP, total phosphorus; TK, total potassium; AN, alkeline–nitrogen; AP, available phosphorus; AK, available potassium; N(L), N content of leaf litter; P(L), P content of leaf litter; K(L), K content of leaf litter; Ca(L), Ca content of leaf litter; Mg(L), Mg content of leaf litter; OC(L), organic carbon content of leaf litter; C:N (L), C:N ratio of leaf litter; C:P(L), C:P ratio of leaf litter; and N:P(L), N:P ratio of leaf litter.

### 3.8. Microbial Network Analysis

Four dominant phyla (Actinobacteria, Proteobacteria, Bacteroidetes, and Firmicutes) were involved in bacterial interactions and three key phyla (Ascomycota, Mortierellomycota, and Basidiomycota) participated in fungal interactions in the three stands (Figure 9a,b). Fertilization decreased the number of collaborative interactions and increased the number of antagonistic interactions of bacteria or fungi (Table S8). The interactions between the microbes showed a similar pattern in the HF and WF forests.

### 3.9. Major Microbes Involved in the Breakdown of Leaf Litter

Fertilization dramatically decreased the abundance of pathogenic and saprophytic fungi but increased the quantity of symbiotic fungi. The abundances of pathogenic and saprophytic fungi in the WF forest were lower than those in the HF forest. The number of symbiotic fungi was greatly increased by WF compared to the CK and HF treatments (Figure S5).

The saprophytic fungi, i.e., *Filobasidium* and *Coprinopsis*, primarily engaged in the breakdown of leaf litter in the unfertilized stands, while *Neocosmospora* mainly contributed to decomposition in the HF and WF forests (Table S9). *Arthrobacter* was the major bacterium involved in litter breakdown in all stands (Table S10).

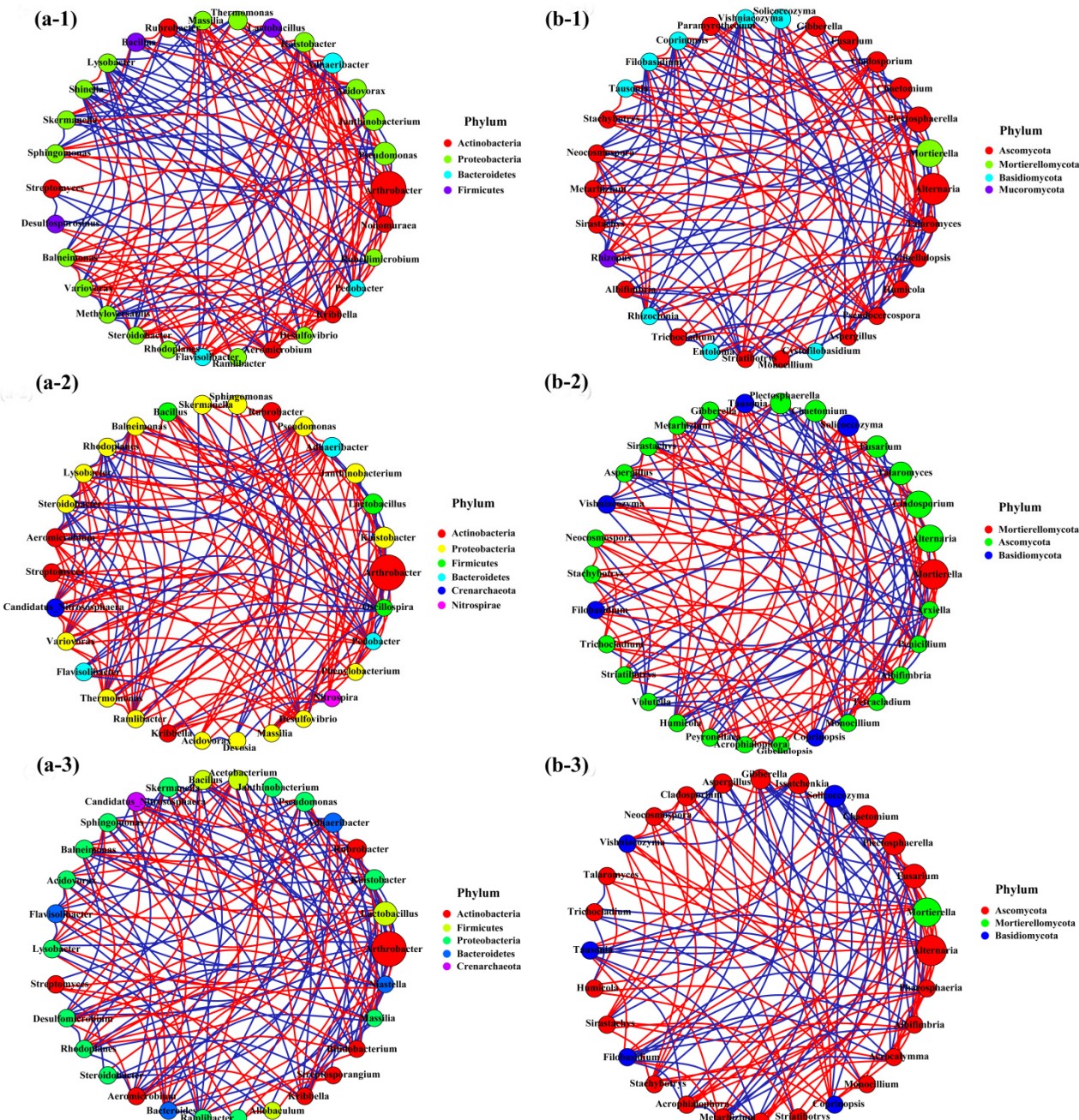

**Figure 9.** Network analysis within microbial communities at the genus level in the three treatments: a, bacteria; b, fungi; (**a-1**)/(**b-1**), CK; (**a-2**)/(**b-2**), HF; and (**a-3**)/(**b-3**), WF. CK, no fertilization; HF, hole fertilization; and WF, integration of water and fertilizer. The different colors reflect different phylum categories, while the circles represent a genus, and each circle's size illustrates the relative abundance. Red lines indicate a positive correlation, and blue lines indicate a negative correlation.

## 4. Discussion

### 4.1. Litter Biomass and C Content in Response to Fertilization

Our results showed that fertilization increased the biomass and C storage of *C. bungei* leaf litter, which was consistent with the report by the author of [47], who found that fertilizer input increased litter production in *Castanopsis sclerophylla* and *Castanopsis eyrei* forests. The reason for this phenomenon was that fertilization boosted the productivity of forests and enhanced litter generation and nutrient return [48,49]. In addition, forest litter production is also affected by climatic conditions [5,7]. In the present study, the biomass and C content in the HF forests were higher than those in the WF forests. However, the dif-

ferences were not easily compared since the same temperatures and levels of precipitation occurred in each stand. The differences may be caused by the quantity and chemistry of the leaves in the various fertilized forests.

The majority of deciduous forests in northern China shed leaves quickly to protect themselves from cold before hibernating (e.g., in November) [50–52]. The seasonal dynamics of the litter biomass in the *C. bungei* plantations demonstrated a single–peak pattern, and the largest litterfall occurred in November. However, bimodal patterns of litter for evergreen broadleaved forests were displayed in subtropical climates. Evergreen species generally grow a large number of new leaves in the rainy season (e.g., in April), increase the shedding of senescent leaves, and experience physiological litterfall in response to water stress in the dry season (e.g., in November) [9]. As indicated in [3], the biomass of leaf litter has been found to be increased during periods of high precipitation in wet seasons. In the present study, however, the amount of litter observed reached a climax regardless of fertilization in the dry season (i.e., October–November), which was higher than that in the wet season. In other words, the litter of *C. bungei* presented higher sensitivity to low temperature rather than precipitation. Thus, the observed differences may have been caused by the leaf anatomy and physiological habits of the evergreen and deciduous trees and the climates of the different geographical locations.

### 4.2. Litter Breakdown in Response to Fertilization

Fertilization greatly decreased the decomposition rate of *C. bungei* leaf litter, which was also confirmed in [26]. However, these results were partially inconsistent with the findings in [53]. In the cited study, the authors discovered that a low–level application of fertilizer accelerated the breakdown of *Pinus armandii* litter but inhibited the process if large amounts of fertilizer were added. Thus, the differences in litter breakdown may be caused by the various fertilization regimes employed and the chemical properties of the litter.

The decomposition rates of litter were found to vary as a function of temperature, moisture, and the quality of litterfall material as indicated by the nutrient concentrations and lignin content in structural tissues [3]. Water–soluble substances and carbohydrates (e.g., soluble sugars and hemicelluloses) are often lost via leaching in the early stages of decomposition (the first 60 days) [54]. In one study, it demonstrated that cellulose and lignin (i.e., the most resistant components) regulated the final stages of litter decomposition [26]. It has been reported that fertilization inhibited the process of the degradation of carbohydrates and lignin [55–59]. In addition, the decomposition rate of litter in the WF forests was slower than that in the HF forests. The differences in litter decomposition may have been caused by the varied chemical composition of the litter (e.g., cellulose and lignin) in the two fertilized stands; however, this was not demonstrated in our current results. Temperature is considered the overriding environmental factor affecting litter decomposition; however, moisture limitation is also recognized as an important factor in many tropical and temperate regions [60]. As described by our results, the decay rate of litter in the wet season (i.e., May–September) presented higher responses to the amount of rainfall and elevated temperature compared to the dry season.

Our study also showed that fertilization reduced nutrient release (e.g., N, P, and K) from leaf litter, indicating that fertilization inhibited the nutrient return from the litter to the soil [27,61,62]. Litter mixtures are thought to support a greater number of microhabitats and increased chemical diversity and could also influence overall decomposition rates and microbial activity through the transfer of nutrients and secondary chemicals [11]. Thus, *C. bungei* forests can be mixed with other tree species to promote litter decomposition and increase nutrient return when fertilizing forests.

### 4.3. Microbial Community in Response to Fertilization

Changes in nutrient inputs can also affect the nutrient statuses of plants, which, in turn, can affect root exudation, leaf litter chemistry, and plant–microbial competition for nutrients. Changes in root biomass can also affect microbial biomass through indirect

effects on the physical soil environment and soil moisture [63]. Long–term excess nutrient deposition can also lead to base cation loss and related changes in soil pH, which can also affect microbial biomass [64,65]. In [66], 8 years of fertilization resulted in a 20%–30% decrease in microbial biomass. Similar research found a decline in soil microbial biomass resulting from chronic elevated N input in forest soils [67]. As indicated by our results, 4–year fertilization had a negative impact on several groups of prevalent microbes (e.g., Proteobacteria and Ascomycota). Most temperate forests in regions with low deposition rates are thought to be nutrient–limited [68,69]. As nutrient limitation is alleviated, plants may reduce their allocation of resources belowground by decreasing root production and exudation, which causes a decline in the quantity of microbial groups.

Proteobacteria are eutrophic organisms with high nutritional needs [70,71]. However, our study indicated that fertilization decreased the abundance of Proteobacteria, which differed from the results presented by the authors of [72]. They believed that eutrophic microbes benefited from fertilizer input. Moreover, the abundance of Proteobacteria in the WF forests was lower than that in the HF forests. The differences in these results may originate from two factors. On the one hand, different fertilization regimes can result in significant differences in microbial communities [19,42,59]. On the other hand, the fast–growing *C. bungei* has a higher demand for soil nutrients when supported by WF [73], resulting in a decrease in soil nutrients (e.g., AN, AP, and AK), which causes an increase in the abundance of oligotrophic bacteria (e.g., Acidobacteria). Furthermore, Firmicutes have been associated with antagonistic activity against phytopathogens [74]. In our study, the number of Firmicutes in the fertilized stands was higher than in the nonfertilized stands, suggesting that the soil environment can be improved via fertilizer input. Ascomycota and Basidiomycota can secrete extracellular enzymes that can breakdown lignin, cellulose, and hemicellulose [75,76]. Our study found that fertilization reduced the quantity of Ascomycota and Basidiomycota, which caused a decrease in litter breakdown.

### 4.4. Key Microbes Involved in Litter Breakdown

In our study, we used correlation relationships between the abundance of OTUs to delineate networks within the analyzed microbial communities. The networks, which were based on random matrix theory in this study, could be robust in terms of their network structure and provide accurate results regarding correlation threshold determination and are thus suitable for studying the modes of biological interaction between systems [77,78]. The network graphs were developed to represent positive and negative relationships between different OTUs and thus, in fact, describe the co–occurrence patterns between OTUs across different samples. Our results indicated that antagonistic interactions between microbes were enhanced in the fertilized soil. The authors of [11] believed that enhanced competition in the microbial community might slow the process of litter decay. Herein, we speculated that the intensive competition between microbes caused by fertilization may not be conducive to litter decomposition, although this was not shown in our present study. Our results also showed that the breakdown rate of litter was not correlated with microbial diversity, which is in agreement with the findings presented in [11]. In the cited study, the authors believed that the nutrient status of the litter, and its organic properties and biotic interactions, seemed to be more important for mass loss values than either microbial richness or community composition.

However, undeniably, some fungi can humify plant residues and degrade complex macromolecular matter into nutrients for plant absorption [79]. The breakdown of litter is mainly aided by saprophytic fungi. In our study, fertilization greatly decreased the quantity of saprotrophs, which resembles the microbial effects seen in Eucalyptus soil [80]. That is, fertilization can prevent leaf litter from decomposing by decreasing the number of saprotrophs. Moreover, the abundance of saprophytic fungi in the HF forests was slightly higher than in the WF forests. The saprophytic fungi were more suited to growth in solid fertilizer environments (i.e., HF) than in environments with liquid fertilizers (i.e., WF) [81]. *Neocosmospora* in the WF and HF forests and *Filobasidium* and *Coprinopsis* in the CK forests

played important roles in litter breakdown. Some bacteria can breakdown the chemical matter of litter. Our results show that fertilization decreased the number of decomposing bacteria, and the number of decomposing bacteria in the HF forests was slightly higher than in WF forests. Surprisingly, *Arthrobacter* was the most prevalent bacteria during the breakdown of leaf litter in all stands.

### 4.5. Limitations and Prospects

Litter decomposition is a long and complicated process, and microbes can undergo a wide range of changes during this process. There were some limitations in our experiments, such as the short duration of litter breakdown. The following areas of study can be addressed in the future. First, the endogenous nutrients of leaf litter (e.g., lignin and cellulose) play a critical role in the breakdown of litter and can be used to monitor the breakdown process. Second, it is necessary to extend the period of litter decomposition, examine the changes in microbial populations at each stage of decomposition, and investigate the microscopic mechanisms using microbial techniques (e.g., genetic inheritance).

### 5. Conclusions

The biomass and C storage of leaf litter in *C. bungei* plantations was enhanced via tree fertilization. Fertilization decreased the breakdown rate of leaf litter mainly by decreasing the abundance of saprophytic fungi and decomposing bacteria. The richness of soil microbes was increased when fertilizer was added. However, the increase in soil microbial diversity did not cause a change in litter decomposition. The integration of water and fertilizer decreased the decomposition rate and nutrient release of leaf litter due to the lower abundance of saprophytic fungi and decomposing bacteria compared to hole fertilization. *Arthrobacter* was the primary bacterium decomposing leaf litter in all forests, and *Neocosmospora* was the main saprophytic fungus responsible for this process in the fertilized environments.

**Supplementary Materials:** The following supporting information can be downloaded at: https://www.mdpi.com/article/10.3390/f14040699/s1, Figure S1: Plot diagrams and arrangement of litter plots, Figure S2: Temperature and precipitation from 2017 to 2022 in Zhangqiu District, Jinan City, China, Figure S3: Stoichiometric ratios of the leaf litter treated through three fertilization regimes. (a), C:N; (b), C:P; (c), N:P; CK, no fertilization; HF, hole fertilization; WF, integration of water and fertilizer. *, $p < 0.05$; **, $p < 0.01$; ***, $p < 0.001$, Figure S4: Linear fitting of bacterial (a,b) and fungal (c,d) diversity to decomposition constants, Figure S5: FUN-Guild analysis for predicting fungal functions in three fertilization regimes. (a) pathotroph; (b) saprotroph; (c) symbiotroph; (d) others. CK: no fertilization; HF: hole fertilization; WF: integration of water and fertilizer, Table S1: Chemical properties of topsoil (0–20 cm) treated through three fertilization regimes, Table S2: Total mass and carbon content of leaf litter in 2021, Table S3: Decomposition constant (*k*) and the duration for leaf litter to decompose by 50% ($t_{50\%}$) or 95% ($t_{95\%}$) in three fertilization regimes, Table S4: Nutrient release rates of leaf litter in three fertilization regimes, Table S5: Relative abundance at the phylum level of bacteria and fungi, Table S6: Alpha diversity of bacteria and fungi in three fertilization regimes, Table S7: The results of envfit function of R package indicated correlations between all parameters and microbial communities (genus level), Table S8: Number of lines in network analysis for bacteria and fungi, Table S9: Relative abundance of saprophytes at genus level in three fertilization regimes, Table S10: Relative abundance of decomposing bacteria at genus level in three fertilization regimes.

**Author Contributions:** Conceptualization, Z.G.; methodology, T.C.; software, D.C.; validation, Y.L.; formal analysis, Q.H. (Qingjun Han); investigation, N.L.; resources, W.M. and J.W.; data curation, Y.S. and J.L.; writing—original draft preparation, Z.G.; writing—review and editing, Z.G.; visualization, Z.G.; supervision, Q.Q. and Q.H. (Qian He); funding acquisition, Q.H. (Qian He). All authors have read and agreed to the published version of the manuscript.

**Funding:** This work was funded by the National Key Research and Development Program of China (2017YFD0600604, 2017YFD060060404).

**Informed Consent Statement:** Informed consent was obtained from all subjects involved in the study.

**Data Availability Statement:** The original contributions presented in the study are included in the article/Supplementary Material. Raw amplicon sequences were deposited in the Sequence Read Archive (SUB12237242 and SUB12237220) and assigned the following BioProject accession numbers: PRJNA896265 and PRJNA896326. Further inquiries can be directed to the corresponding authors.

**Conflicts of Interest:** There are no conflict of interest to declare.

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
