# Peer review of "Leaf Litter Breakdown and Soil Microbes in Catalpa bungei Plantations in Response to Various Fertilization Regimes"

_forests, doi:10.3390/f14040699_

Round 1

Reviewer 1 Report

Many recent and prior publications on the topic were omitted. Authors’ proposition that the question is studied insufficiently is not so correct.

Author Response

Response to Reviewer 1 Comments:

Point 1: L15: Doubtful proposition. There are many publications....

Response 1: The statement is revised and please refer to line 14-16.

 Point 2: L40-41: only single source? There are a lot of valuable publications

Response 2: The quantity of references are supplemented and please refer to line 39-41.

 Point 3: L127-128, 134: how?

Response 3: The specific statements are addressed and please refer to line 141-144, 149-151.

Point 4: If one of the referees has suggested that your manuscript should undergo extensive English revisions, please address this issue during revision.

Response 4: The English editing service of MDPI is used, and the English editing ID is english-62722. The English editor, whose name is Robert John MORAN, in charge of editing the manuscript.

Reviewer 2 Report

This is one of those rare studies that analyzes the effect of fertilizer application on forest stands. For which it is very much appreciated.

L4: Please fix the title.

L39: This sentence reads very strange.

L40-41: “forestland” ? All vegetative ecosystems depend on litter nutrient cycling.

L39-42: The authors refer in general to litter comprising leaves, branches, wood, miscellaneous. Therefore, the role of insects should be mentioned: https://doi.org/10.1038/s41586-021-03740-8

L50-63: Why is it not mentioned that decomposition as well as litterfall depends on climatic conditions, type of ecosystem, type of species (evergreen or deciduous), litter quality (lignin and cellulose), etc.?

L111: Why were these fertilizers selected and in these concentrations?

L121: “S” shape? The paper would benefit from including an image of the study area and photographs of the experimental designs for litter collection and decomposition.

L129-130: a pore size of 40 mesh? This is not clear. Moreover, the methodology is very similar to the one used in this study and could be used as a reference: https://doi.org/10.1659/MRD-JOURNAL-D-16-00004.1

L137-143: The soil samples were taken in free areas of the forest? or were they taken near the trunks of the trees, it is possible that there are anomalous results because the areas near the tree are enriched by stemflow with respect to the areas distant from the trunk, check: https://doi.org/10.1002/eco.2348

L149-150: MS-1, MS-2, MS-3: I think they are important parts that should be part of the paper.

L237-243: I am not very clear about the patterns that the authors observe in the figures, if they can describe it in a better way it would be very helpful for the reader.

Figure 7: Why are there black and white asterisks?

Figure 9: The names are not legible, although I can't think of a different way to present them.

L341-355: It would be interesting to compare leaf litterfall and decompisicon results in contrasting environments such as wet (https://doi.org/10.1659/MRD-JOURNAL-D-16-00004.1) and dry (https://doi.org/10.1093/forestry/74.4.347).

L396-403: I like this idea, but it is highly speculative, and a warning should be issued to take this statement with caution.

I would like to draw the authors' attention to some studies that would be useful to complement and compare results:

https://doi.org/10.1007/978-3-319-27422-5_13

https://doi.org/10.1007/s00442-012-2403-z

https://doi.org/10.1007/s10021-011-9516-9

https://doi.org/10.1016/j.foreco.2005.11.002

Author Response

Response to Reviewer 2 Comments:

Point 1: L4: Please fix the title.

Response 1: The title is fixed and please refer to line 2-3.

Point 2: L39: This sentence reads very strange.

Response 2: This sentence is revised and please refer to line 39-40.

Point 3: L40-41: “forestland” ? All vegetative ecosystems depend on litter nutrient cycling.

Response 3: This statement is revised and please refer to line 40-41.

Point 4: L39-42: The authors refer in general to litter comprising leaves, branches, wood, miscellaneous. Therefore, the role of insects should be mentioned: https://doi.org/10.1038/s41586-021-03740-8

Response 4: The relevant information is addressed and please refer to line 41-43.

Point 5: L50-63: Why is it not mentioned that decomposition as well as litterfall depends on climatic conditions, type of ecosystem, type of species (evergreen or deciduous), litter quality (lignin and cellulose), etc.?

Response 5: The relevant information is addressed and please refer to line 41-43.

Point 6: L111: Why were these fertilizers selected and in these concentrations?

Response 6: In the past, several fertilization amounts were tried for Catalpa bungei plantations. And the optimum regime for hole fertilization, i.e., N (24 g/tree), P2O5 (8 g/tree), and K2O (16 g/tree), is selected for Catalpa bungei fertilized-plantations, which is not published. Here, the fertilization scheme, i.e., integration of water and fertilizer used widely in the agricultural ecosystems, is consulted and held the same amount of fertilizer as hole fertilization.

Point 7: L121: “S” shape? The paper would benefit from including an image of the study area and photographs of the experimental designs for litter collection and decomposition.

Response 7: The diagram of experimental designs and litter box is added in Figure S1.

Point 8: L129-130: a pore size of 40 mesh? This is not clear. Moreover, the methodology is very similar to the one used in this study and could be used as a reference: https://doi.org/10.1659/MRD-JOURNAL-D-16-00004.1

Response 8: The statement is revised and please refer to line 133-134, 143-144.

Point 9: L137-143: The soil samples were taken in free areas of the forest? or were they taken near the trunks of the trees, it is possible that there are anomalous results because the areas near the tree are enriched by stemflow with respect to the areas distant from the trunk, check: https://doi.org/10.1002/eco.2348

Response 9: The statement is revised and please refer to line 154-155.

Point 10: L149-150: MS-1, MS-2, MS-3: I think they are important parts that should be part of the paper.

Response 10: The information is added and please refer to line 120-130, 165-177, 180-213.

Point 11: L237-243: I am not very clear about the patterns that the authors observe in the figures, if they can describe it in a better way it would be very helpful for the reader.

Response 11: First, the response of six nutrition on fertilization methods are intended to be explored. Thus, the mean of total release rates during different decay period of litter, such as 60d, 120d, 180d, 240d, and 300d, are determined to compared the effects of fertilization methods on release rates, which are displayed in Table S4. Second, the trends of nutrient release are intended to be described in the figure 4. Please refer to line 295-301.

Point 12: Figure 7: Why are there black and white asterisks?

Response 12: The color of asterisks are revised and please refer to Figure 7.

Point 13: Figure 9: The names are not legible, although I can't think of a different way to present them.

Response 13: Ultra-high definition version of original image is uploaded to the editor in forms of pictures.

Point 14: L341-355: It would be interesting to compare leaf litterfall and decompisicon results in contrasting environments such as wet (https://doi.org/10.1659/MRD-JOURNAL-D-16-00004.1) and dry (https://doi.org/10.1093/forestry/74.4.347).

Response 14: The information is added and please refer to line 412-417, 437-442.

 Point 15: L396-403: I like this idea, but it is highly speculative, and a warning should be issued to take this statement with caution.

Response 15: The explanations based on the previous results are speculative, but the euphemistic statements are elaborated. And please refer to line 483-499.

Round 2

Reviewer 1 Report

authors have considered previous comments. The manuscript could be accepted. 

Author Response

Point 1: authors have considered previous comments. The manuscript could be accepted. 

Response 1: Thanks to the statements from reviewer.

Reviewer 2 Report

The authors have responded adequately to all my comments. After revising the article, a great improvement is noticeable.

Two last observations:

1) Mention should be made of what was done to determine these fertilizers and their concentrations and say that the information has not been published. (Point 6: L111: Why were these fertilizers selected and in these concentrations? Response 6: In the past, several fertilization amounts were tried for Catalpa bungei plantations. And the optimum regime for hole fertilization, i.e., N (24 g/tree), P2O5 (8 g/tree), and K2O (16 g/tree), is selected for Catalpa bungei fertilized-plantations, which is not published. Here, the fertilization scheme, i.e., integration of water and fertilizer used widely in the agricultural ecosystems, is consulted and held the same amount of fertilizer as hole fertilization.)

2) Figure S1 was not added, the document starts with figure S2 (meteorological data). Please add Figure S1 (study area and sampling methodology).

Author Response

Point 1: Mention should be made of what was done to determine these fertilizers and their concentrations and say that the information has not been published. (Point 6: L111: Why were these fertilizers selected and in these concentrations? Response 6: In the past, several fertilization amounts were tried for Catalpa bungei plantations. And the optimum regime for hole fertilization, i.e., N (24 g/tree), P2O5 (8 g/tree), and K2O (16 g/tree), is selected for Catalpa bungei fertilized-plantations, which is not published. Here, the fertilization scheme, i.e., integration of water and fertilizer used widely in the agricultural ecosystems, is consulted and held the same amount of fertilizer as hole fertilization.)

Response 1: The statement is revised and please refer to line 117-122.

Point 2: Figure S1 was not added, the document starts with figure S2 (meteorological data). Please add Figure S1 (study area and sampling methodology).

Response 2: The statement is revised and please refer to line 107 and Figure S1 from Supplementary materials.
